# Intraoperative Neuromonitoring for Thyroid Surgery in Children and Adolescents: A Single Center Experience

**DOI:** 10.3390/children9121992

**Published:** 2022-12-18

**Authors:** Cristina Martucci, Silvia Madafferi, Alessandro Crocoli, Franco Randi, Erika Malara, Viviana Ponzo, Maria Debora De Pasquale, Alessandro Inserra

**Affiliations:** 1General Surgery Unit, Department of Surgery, Bambino Gesù Children’s Hospital—IRCCS, 00165 Rome, Italy; 2Surgical Oncology Unit, Department of Surgery, Bambino Gesù Children’s Hospital—IRCCS, 00165 Rome, Italy; 3Neurosurgery Unit, Department of Neuroscience and Psychiatry Sciences, Bambino Gesù Children’s Hospital—IRCC, 00165 Rome, Italy; 4Hematology Oncology Unit, Department of Pediatric Hematology/Oncology Cell and Gene Therapy, Bambino Gesù Children’s Hospital—IRCCS, 00165 Rome, Italy

**Keywords:** thyroid cancer surgery, thyroid cancer, thyroid cancer treatment, nerve integrity monitoring, children

## Abstract

Intraoperative neuromonitoring (IONM) of the recurrent laryngeal nerve (RLN) has been shown in adults to minimize nerve palsy after thyroid surgery, but only few studies on its efficacy in a pediatric population have been reported. We conducted a retrospective study on patients operated for thyroid lesions from 2016 to 2022. The analyzed population was divided in two groups: patients treated from 2016 to 2020, when the identification of the RLN was performed without IONM (Group A); and patients treated since 2021, when IONM was implemented in every surgical procedure on the thyroid (Group B). Intraoperative Neurophysiological Monitoring was performed by using corticobulbar motor-evoked potentials and continuous electromyography. Twentyfive children underwent thyroid resection, 19 (76%) of which due to thyroid carcinoma. Each patient’s recurrent nerve was identified; IONM was used in 13 patients. In Group A, one temporary nerve palsy was identified postoperatively (8.3%), while in group B one nerve dysfunction occurred (7.7%). No statistically significant difference was found between the two groups in terms of post-operative RLN palsy. No surgical complication due to the use of IONM was reported. In children and teenagers, intraoperative neuromonitoring of the recurrent laryngeal nerve is a safe and accurate method, minimizing the risk of nerve damage.

## 1. Introduction

The importance of identifying the recurrent laryngeal nerve (RLN) during thyroid surgery is well established [1,2,3]. The benefit of intraoperative neuromonitoring (IONM) of the RLN over macroscopic RLN identification without IONM has recently been demonstrated, particularly in high-risk operations such as those for Grave’s disease, recurrent goiter, or thyroid carcinomas [4,5,6,7]. Little data have been reported in the literature on the use of IONM in children and adolescents [8,9]. Regarding thyroid surgery, there are anatomical and physiological differences between children and adults, such as trachea, larynx, and RLN’s smaller diameter, suggest further benefits of IONM. This retrospective study analyzed children and adolescents who underwent thyroid surgery in our department, with versus without IONM, to determine its utility and potential benefits.

## 2. Materials and Methods

IONM was performed using corticobulbar motor-evoked potentials (CoMEPs) and continuous electromyography (cEMG). CoMEPs were obtained using an electric stimulation (TES) delivering a stimulus pulse train of 4–7 pulses (single pulse width: 0.5–0.75 µsec), with an interstimulus interval (ISI) of 4 msec. According to the International 10–20 system, the stimulating electrodes were positioned at C3, C4, and Cz with bipolar montage. CoMEPs were tested every 15 min from baseline registration until the end of the surgery.

CoMEPs and cEMG were recorded from the RLN (32-channel system; Cadwell Elite, Kennewick, WA, USA). All patients were treated under general anesthesia and intubated with a Nerve Integrity Monitor (NIM) Standard Reinforced EMG Endotracheal Tube (Medtronic Xomed Inc., Dublin, Ireland; 6.0–8.0 mm I.D.). Electrode impedance was routinely confirmed as <5 kΩ before the surgical incision was made.

EMG recordings were set at a gain of 50–200 mV, filter setting of 30–5000 Hz, and sweep length of 200 msec per division.

We retrospectively analyzed all children and adolescents who underwent surgery for thyroid diseases by the same surgical team between January 2016 and June 2022. Informed consent was obtained from the patients or their parents.

The analyzed population was divided into two groups: Group A, patients who underwent surgery between 2016 and 2020 and in whom the RLN was identified without IONM; and Group B, patients who underwent surgery during or after 2021, when we started using IONM.

We examined the number of RLN lesions in both groups, by patient age, underlying thyroid disease, and resection extent. Complications with the IONM including bleeding into the vocal cords, orotracheal tube cuff damage, and postoperative cervical infections were also documented. 

Differences were calculated using the chi-squared (X^2^) test in GraphPad Prism Software (version 8.4.3; GraphPad, San Diego, CA, USA). *p* values of <0.05 were considered statistically significant.

## 3. Results

Since 2016, 25 children underwent thyroid surgery at our institution. Among them, in 12 patients, who underwent surgery between 2016 and 2020, the RLN was identified without IONM (Group A); the other 13 underwent surgery since 2021, in which the RLN was identified with IONM (Group B).

The median age was 13 years (range 3.9–18 years) for Group A and 12.8 years (range 1.5–13.9 years) for Group B. Sixteen patients (64%) were female; of them, six were in Group A (50%) and ten were in Group B (76.9%).

Most of the procedures were performed to treat thyroid cancer: 17 patients presented with papillary carcinoma (8 in Group A, 9 in Group B), 1 with follicular carcinoma (in Group A), and 1 with medullary carcinoma (in Group B). Moreover, in 3 patients in Group A, thyroid adenoma was confirmed; in Group B, 3 thyroidectomies were performed for symptomatic Hashimoto thyroiditis (n = 1) and nodular thyroid hyperplasia (n = 2).

Due to the extent of the thyroid disease, the resection ranged from hemithyroidectomy (n = 3 in Group A, n = 4 in Group B) to total thyroidectomy (n = 9 in each group) via Kocher access.

There were no major intergroup differences in age or resection extent.

Table 1 summarizes the patients’ clinical characteristics, pathologies, and surgical approaches.

In all the patients, bilateral RLNs were clearly identified in the surgical field and preserved; nevertheless, there was one case of temporary nerve palsy in Group A (8.3%) versus one case of nerve dysfunction in Group B (7.7%).

The patient in Group A was a 13.5-year-old boy who underwent total thyroidectomy for papillary carcinoma and presented with temporary dysphonia and bitonal voice in the immediate post-operative period that spontaneously resolved a few hours later. No nerve impairment was reported during follow-up.

The patient in Group B was a 1.5-year-old boy (the youngest patient in the study) with multiple endocrine neoplasia type 2B. In this case, a total thyroidectomy was performed, and cEMG activation of the laryngeal musculature with a reduction in CoMEP at the end versus beginning of the intervention was noted. A pathological examination confirmed the presence of medullary microcarcinoma in the bilateral thyroid lobes without neoplastic involvement of the lymph nodes. After the procedure, the patient presented with dysphonia, hoarseness, and wheezing; on postoperative day 5, laryngoscopy revealed a hypomobile left vocal cord with glottic insufficiency during phonation. At the 1-year follow-up, the symptoms had significantly improved, although laryngoscopy showed no significant difference.

The differences in RLN palsy between children who underwent surgery with and without IONM were not statistically significant according to the X^2^ test (*p* = 0.9529).

No complications caused by IONM were reported, such as cuff injury, hematoma of the vocal cords, or postoperative cervical infection, while two cases of post-operative hematoma (n = 1 in each group) occurred.

## 4. Discussion

Transient or permanent RLN lesions, hypoparathyroidism, postoperative infection, and bleeding are possible complications of thyroid surgery [10]. RLN lesions may result from transection, clamping, stretching, electro-thermal injury, ligature entrapment, or ischemia during the surgical procedure [11]. This complication in children and adolescents has a reported frequency of 0.5–10% for permanent lesions and 2–28% for temporary lesions, depending on the underlying thyroid disease and resection extent [8,9,10,12,13]. The implications of RLN lesions, such as voice problems and dyspnea (possibly requiring tracheostomy), are feared, especially in children.

Many studies have demonstrated the importance of identifying the RLN during thyroid surgery in adults, particularly in cases of thyroid carcinomas, recurrent goiter, Graves’ disease, or an abnormal RLN course [2,4,14].

Malignant pathologies, secondary operations, and re-operations for hemorrhage, non-identification of the nerve, anatomic variability (e.g., extra-laryngeal branching) or anomaly (e.g., non-RLN), and anatomic distortion from goiter or tumor are factors that increase the risk of RLN injury [15]. Differently from adults, children are more vulnerable to surgically induced injuries due to the pediatric population’s smaller trachea, larynx, vessels, and nerves, anatomical variations of the vascular and nervous course (especially in cases of RLN), the potential presence of the thymus in the caudal cervical compartment, and the size of the pathological thyroid compared to the normal thyroid. 

Several approaches have been developed for detecting the RLN during thyroid surgery. IONM is a recent approach adopted in general practice and notably in pediatric surgery [9,16], with a reported high negative predictive value of 92–100% but a low and highly variable positive predictive value of 10–90% [1,17]. Many IONM techniques have been developed for identifying and preserving the RLN during thyroid surgery [9,16], but only a few studies have been reported about the benefits of IONM in children and adolescents [8,9].

In the current study, we applied CoMEP from the laryngeal muscles in addition to free-run EMG. CoMEP is not frequently used in thyroid surgery, but it has been routinely adopted in IONM of the cranial nerves in posterior cranial fossa surgery [18]. Our results demonstrated that the signals generated by these techniques were stable and repeatable, even in challenging anatomical and anesthetic settings like the pediatric one. We noted no IONM-related complications, such as postoperative infections, cuff injuries, or hematoma of the vocal cords, proving that this procedure is safe even for younger patients.

Prior researches, especially in adults, showed that IONM combined with visual RLN identification considerably reduces postoperative RLN palsy rates [4,8,9,19,20,21,22], without lengthening the surgical procedure [23,24]. Especially in complicated operations, such as re-operations, residual thyroidectomy, or inflammation, the RLN may be encased within or displaced by scar tissue, making its visual identification challenging and time-consuming, even for experienced surgeons. In these situations, IONM offers a clear advantage for identifying the RLN and its topographic course [14]. Another benefit of IONM during thyroid surgery is that it offers immediate audible and visual feedback, enabling adjustments to the surgical strategy as needed. The assessment of nerve integrity at the conclusion of the resection is also made possible by nerve monitoring [9].

Although the intergroup differences were not statistically significant, our data suggest that IONM may help reduce the number of nerve lesions. Because of the small number of patients in our study (due to the rarity of thyroid disease in the pediatric population) as well as the overall low incidence of complications, the benefit of IONM in children and adolescents was not statistically confirmed; however, the incidence of RLN palsy was slightly lower in Group B (7.7%), in which IONM was used, compared to Group A (8.3%). Moreover, the patient from Group B with the RLN impairment presented very peculiar clinical characteristics: he was much younger (1.5 years old) than all the other children in our study (median age: 13 years for Group A and 12.8 years for Group B) and cases reported to date [8,9], predisposing him to the creation of incidental lesions of microscopic vascular or nervous structures during surgical maneuvers (by accidental touch or electrocautery). In this patient, cEMG activation of the laryngeal musculature and a decrease in CoMEP after the conclusion of the intervention versus baseline were noted. Considering the medium-term recovery of function in this patient, CoMEPs might function as a helpful predictive index.

However, IONM showed no significant variation during the procedure, suggesting the need to create a new variation of this technique as well as tailor-made tools for younger patients, who are more sensitive to minimal changes in pathways and have smaller anatomical features. Furthermore, IONM is quite sensitive but not very specific in this setting. During the procedure, various EMG activations from the muscles that were investigated were recorded, in connection to both direct manipulations and indirect solicitation (at risk of injury). In the latter scenarios, the operator should alert the neurophysiologist if he suspects that significant structures are being directly involved (without waiting for the monitoring to give warning signals), in order to correctly interpret the data. Therefore, to properly use and analyze the signals produced by IONM, correct and constant communication between operators is mandatory.

## 5. Conclusions

Recent evidence from thyroid surgery conducted in adults has demonstrated the benefits of IONM in RLN identification and its postoperative preservation. Because of the low morbidity of IONM, even in the pediatric population, we conclude that IONM during thyroid surgery in children and adolescents is of additional benefit.

Owing to the rarity of thyroid procedures performed in children and adolescents, even in specialized surgical centers, a prospective multicenter study including a large number of patients is mandatory to demonstrate a statistically significant advantage of IONM in this population.

## Figures and Tables

**Table 1 children-09-01992-t001:** Patients’ Demographics, Surgical Procedure and IONM outcome.

	Group A	Group B	*p* Value (X^2^ Test)
Patients (n.)	12	13	
*Female*	*6*	*10*	
*Male*	*6*	*3*	
Median Age (Years)	13 (3.9–18)	12.8 (1.5–13.9)	
Indication for Surgery			
*Papillary Thyroid Carcinoma*	*8*	*9*	
*Follicular Thyroid Carcinoma*	*1*	*0*	
*Medullary Thyroid Carcinoma*	*0*	*1*	
*Nodular Thyroid Hyperplasia*	*0*	*2*	
*Hashimoto Thyroiditis*	*0*	*1*	
*Adenoma*	*3*	*0*	
Extension of Resection			
*Total Thyroidectomy*	*9*	*9*	
*Hemithyroidectomy*	*3*	*4*	
Post-operative RLN Palsies n (%)	1 (8.3%)	1 (7.7%)	0.9529
Other Surgical Complications (n/%)	1 (8.3%) *	1 (7.7%) *	0.9529

* = Post-operative bleeding.

## Data Availability

The data presented in this study are available on request from the corresponding author. The data are not publicly available due to privacy reasons.

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
