# Peer review of "Intraoperative Neuromonitoring for Thyroid Surgery in Children and Adolescents: A Single Center Experience"

_children, 2022, doi:10.3390/children9121992_

Round 1
Reviewer 1 Report
CoMEP use is now standard of care in most pediatric oncologic institutions.
(1) The Discussion part would benefit from adding literature references on the improved morbidity without impacting operative time and workflow.
(2) Discuss potential costs related to the use of CoMEP.
(3) Currently, your data do not support your discussion. There are no statically significant differences between groups. Revise your discussion accordingly and strengthen the potential advantages of using IONM.
Author Response
The Discussion part would benefit from adding literature references on the improved morbidity without impacting operative time and workflow.
We added some references about this topic, as suggested, in the discussion paragraph (Princi 2022, Fei 2022)
Discuss potential costs related to the use of CoMEP.
Unluckily, there’s no specific literature about the costs related to the use of CoMEP only. Furthermore, literature is equivocal on the issue of cost-effectiveness of IONM with studies both for and against, depending on the various methods of cost-effectiveness analysis utilised and variations in assumptions in each model.
Using the Markov model, l-Qurayshi et al. hypothesized that IONM is more cost-effective than standard visual examination, as reported by Wong et al through simulation economic modeling by. Contrary to what was previously stated, Rocke’s analysis using a decision tree model came to the conclusion that IONM would only be cost-effective if the surgeon could reduce his nerve palsy rate by more than 50.4% using IONM as compared to visual inspection, implying an erroneously high palsy rate. Using a related decision tree model, Sanabria et al. discovered that the incidence of nerve palsy was the same with and without IONM (1% vs. 1.6%). The type of healthcare system (universal or not), the typical cost of surgery, and patient preferences are other variables that depend on a country's socioeconomic level and geographic location. The direct and indirect costs of therapy, litigation, phonosurgery, and recovery time are all significant and must be taken into consideration.
Some references about this topic are:
- Al-Qurayshi Z, Kandil E, Randolph GW (2017) Cost effectiveness of intraoperative nerve monitoring in avoidance of bilateral recur- rent laryngeal nerve injury in patients undergoing total thyroidecto- my. Br J Surg 104:1523–1531. https://doi.org/10.1002/bjs.10582
- Al-Qurayshi Z, Robins R, Hauch A, Randolph GW, Kandil E (2016) Association of surgeon volume with outcomes and cost savings following thyroidectomy: a national forecast. JAMA Otolaryngol Head Neck Surg 142(1):32–39. https://doi.org/10. 1001/jamaoto.2015.2503
- Wang T, Kim HY, Wu CW, Rausei S, Sun H, Pergolizzi FP, Dionigi G (2017) Analyzing cost effectiveness of neural- monitoring in recurrent laryngeal nerve recovery course in thyroid surgery. Int J Surg 48:180–188. https://doi.org/10.1016/j.ijsu.2017. 10.003
- Rocke DJ, Goldstein DP, de Almeida JR (2016) A cost-utility anal- ysis of recurrent laryngeal nerve monitoring in the setting of total thyroidectomy. JAMA Otolaryngol Head Neck Surg 142:1199– 1205. https://doi.org/10.1001/jamaoto.2016.2860
- Sanabria Á, Ramírez A (2015) Economic analysis of routine neuromonitoring of recurrent laryngeal nerve in total thyroidectomy. Biomedica 35(3):363–371. https://doi.org/10. 7705/biomedica.v35i3.2371
- Deshmukh A, Thomas AE, Dhar H, Velayutham P, Pantvaidya G, Pai P, Chaukar D (2022) Seeing Is Not Believing: Intraoperative Nerve Monitoring (IONM) in the Thyroid Surgery. Indian J Surg Oncol Mar;13(1):121-132. doi: 10.1007/s13193-021-01348-y.
In our population, no significative differences in terms of costs have been reported between population A and B.
Currently, your data do not support your discussion. There are no statically significant differences between groups. Revise your discussion accordingly and strengthen the potential advantages of using IONM.
We revised the discussion according to your suggestion.
Reviewer 2 Report
Interesting comparative study on neuromonitoring of the recurrent laryngeal nerve in pediatric thyroid surgery; the benefits of neuromonitoring in thyroid surgery are known and, considering the limited data available in the literature on thyroid surgery in children, the small number of patients included in the study can be considered useful for increasing experience in this area.
Allow me to suggest the inclusion among the bibliographic entries in the discussion section of recent work on the subject published in a prestigious journal that could increase the quality of this paper: Zhang D, Sun H, Kim HY, Pino A, Patroniti S, Frattini F, Impellizzeri P, Romeo C, Randolph GW, Wu CW, Dionigi G, Fama' F. Optimal Monitoring Technology for Pediatric Thyroidectomy. Cancers (Basel). 2022 May 24;14(11):2586. doi:10.3390/cancers14112586.
English should be carefully reviewed and improved.
Therefore, overall, I recommend acceptance after minor, mostly linguistic, revision.
Author Response
Interesting comparative study on neuromonitoring of the recurrent laryngeal nerve in pediatric thyroid surgery; the benefits of neuromonitoring in thyroid surgery are known and, considering the limited data available in the literature on thyroid surgery in children, the small number of patients included in the study can be considered useful for increasing experience in this area.
Thank you for your valuable comments.
Allow me to suggest the inclusion among the bibliographic entries in the discussion section of recent work on the subject published in a prestigious journal that could increase the quality of this paper: Zhang D, Sun H, Kim HY, Pino A, Patroniti S, Frattini F, Impellizzeri P, Romeo C, Randolph GW, Wu CW, Dionigi G, Fama' F. Optimal Monitoring Technology for Pediatric Thyroidectomy. Cancers (Basel). 2022 May 24;14(11):2586. doi:10.3390/cancers14112586.
We added this reference, as suggested, in the discussion paragraph
English should be carefully reviewed and improved.
We made a complete English revision of the manuscript
Therefore, overall, I recommend acceptance after minor, mostly linguistic, revision.